# 1D axial heterostructure of hydrogen-bonded framework and metal-organic framework by metalation reaction

Siquan Zhang [1], Yong-Sheng Wei [2] ✉, Ellan K. Berdichevsky[1], Loris Lombardo [3], Zeyu Fan [1], Cheng Luo[1], Masahiko Tsujimoto [2], Nao Horike[2] & Satoshi Horike [2,3,4] ✉

Creating one-dimensional (1D) axial heterostructures of crystals formed from hydrogen-bonded organic frameworks (HOFs) and metal-organic frameworks (MOFs) is challenging due to their distinct chemical bonds to construct each porous architecture. In this study, we applied the metalation process, exchanging $H^+$ with monovalent metal cations, to fabricate bulk crystal 1D axial heterostructures of HOF and MOF. Rod-shaped single crystals of HOFs with [N−H···N] hydrogen bonds were immersed in a $Cu^+$ solution to induce metalation, resulting in MOF | HOF | MOF crystals exhibiting a 1D axial heterostructure. X-ray diffraction, scanning electron microscopy, gas sorption, and emission microscopy demonstrated that metalation began at both ends of the rod-shaped HOF crystals and took 48 hours to transform into MOF. The spatially and temporally controlled metalation facilitated the regulation of the sizes of HOF and MOF domains in the 1D axial heterostructure. The MOF | HOF | MOF crystals exhibited interface-controlled gas diffusion and various spatially-resolved photoluminescence behaviors depending on the distribution of each component.

Heterostructure crystals are important for the study of hierarchical physical/chemical properties and interface functionalities[1]. One of the most notable types is the one-dimensional (1D) axial heterostructure[2,3]. The unique features and advantages of the 1D axial heterostructures compared with other heterostructures (core-shell, superlattice, graded, and Janus structures) are mainly based on their highly anisotropic (1D) crystal arrays, which are usually produced by sequential growth using parent crystals[4,5]. 1D axial heterostructure crystals of organics[6–8], pnictides[9–11], and chalcogenides[12,13] provide unique properties such as multiple spatial-resolved emissions and tunable axial transport of charge carriers[14]. This provides potential applications such as information storage[15] and axial carrier transport for solar energy

conversion[16]. Such properties are not feasible in single-phase crystals, indicating the significance of the heterostructures. This also applies to porous molecular frameworks, including metal-organic frameworks (MOFs)[17–22], covalent-organic frameworks (COFs)[23–26], and hydrogen-bonded organic frameworks (HOFs)[27–30]. However, creating 1D axial heterostructures from these framework systems is difficult because of the differences in the characteristics of chemical bonding[31–34]. For example, there are the reported heterostructures of HOF and MOF crystals, but none of them belong to the 1D axial heterostructure[35,36]. It has been a challenge to create 1D axial heterostructures of HOF and MOF by controlling the formation of both hydrogen bonds and coordination bonds.

[1]Department of Synthetic Chemistry and Biological Chemistry, Graduate School of Engineering, Kyoto University, Kyoto 615-8510, Japan. [2]Institute for Integrated Cell-Material Sciences, Institute for Advanced Study, Kyoto University, Yoshida-Honmachi, Sakyo-ku, Kyoto 606-8501, Japan. [3]Department of Chemistry, Graduate School of Science, Kyoto University, Kitashirakawa-Oiwakecho, Kyoto 606-8502, Japan. [4]Department of Materials Science and Engineering, School of Molecular Science and Engineering, Vidyasirimedhi Institute of Science and Technology, Rayong 21210, Thailand. ✉e-mail: wei.yongsheng.3t@kyoto-u.ac.jp; horike.satoshi.3r@kyoto-u.ac.jp

On metallic surfaces, it is known that the single-layer HOF to MOF with topotactic transformation occurs through the metalation process[37,38]. The protons (H[+]) in the single-layer HOF react with the metal atoms on the surface to form metal ions, which participate in the construction of the MOFs with a high conversion rate. We had the idea that we could create 1D axial heterostructures of HOF and MOF in bulk crystals by utilizing the metalation process.

In this study, two porous pyrazole-based HOFs were synthesized to achieve this purpose. The rod-shaped HOF single crystals were prepared and reacted with Cu[+] solutions to promote metalation, resulting in 1D axial heterostructures of HOF and MOF. Thermogravimetric analyses demonstrated that the metalation occurs stoichiometrically for 2 days at 298 K. Controlled metalation results in the formation of MOF | HOF | MOF crystals with a 1D axial heterostructure. Time-course X-ray diffraction and scanning electron microscopy measurements revealed the growth mechanisms and spatial distribution of HOF and MOF domains within the heterostructure. The MOF | HOF | MOF heterostructures exhibit tunable BET surface areas and stimuli-responsive photoluminescence based on the spatial distribution of each component. This approach provides precise control over the size and distribution of HOF and MOF domains by overcoming different chemical bonds and controlling the metalation process.

## Results and discussion
### Synthesis and structural characterization of HOF-a and HOF-b
One-step Schiff-base reaction was employed to synthesize (benzene-1,3,5-triyl)-tris(1-(3,5-dimethyl-1H-pyrazol-4-yl)-methanimine (**H₃L1**, Fig. 1a and Supplementary Figs. 1–3) and (1,3,5-triazine-2,4,6-triyl)-tris(benzene-4,1-diyl))-tris(1-(3,5-dimethyl-1H-pyrazol-4-yl)-methanimine (**H₃L2**, Fig. 1b and Supplementary Figs. 4–6). Crystal structure

analysis of **H₃L1** by single-crystal X-ray diffraction (SCXRD) confirmed the hydrogen-bonded network (Fig. 1a and Supplementary Fig. 7). We denote it as **HOF-a**. Figure 1c shows that three pyrazoles form a triplet via three [N–H···N] hydrogen bonds (2.893 Å of N···N distance, NHN angles of 164.2°). The steric hindrance caused by the dimethyl group makes the three pyrazole rings non-coplanar (Supplementary Fig. 8). The dihedral angle of the pyrazole rings is 11.5° between the central benzene rings. Each **H₃L1** connects with three different triplets, forming a two-dimensional layer with a 3,3′-connected **hnb** topology. The layers are stacked together in an AB-staggered fashion through the π···π interaction between pyrazoles and the benzene ring from adjacent layers, forming a framework with 1D channels of 0.8 nm in diameter along the *c*-axis.

The structure model of **HOF-b** was created from **HOF-a** and optimized based on the unit cell refined by powder X-ray diffraction (PXRD) of **HOF-b** (Fig. 1b)[39–41]. The calculated PXRD pattern from the model agrees with the experimental one (Supplementary Fig. 9). **HOF-b** has a pore diameter of 1.7 nm (Supplementary Fig. 10). The N₂ sorption isotherms at 77 K of activated **HOF-a** and **HOF-b** showed the Type-I[42] profiles (Supplementary Figs. 11 and 12). The Brunauer-Emmett-Teller (BET) surface areas of **HOF-a** and **HOF-b** were 1956 and 2105 m² g⁻¹, respectively. The pore sizes estimated by a non-local density functional theory (NLDFT) kernel were 0.7 nm and 1.6 nm, which match well with those from crystal structures.

### Metalation of HOF-a and HOF-b
Soaking the colorless crystals of **HOF-b** in a methanol solution of Cu(CF₃SO₃) for 48 hours at 298 K gave yellow crystals. The crystals were washed three times with methanol to remove unreacted Cu(CF₃SO₃), and named **MOF-b-Cu** (Fig. 2a, Supplementary Figs. 13 and 14, and

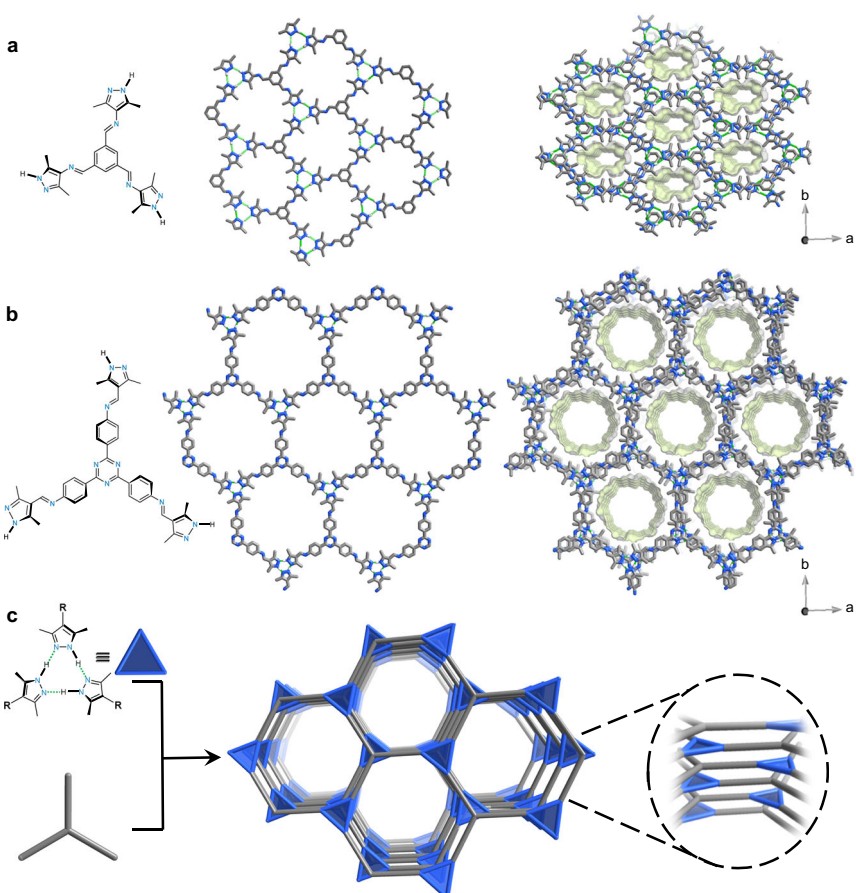

**Fig. 1 | Crystal structures of the HOFs.** Crystal structures of single layer and stacking layers of (**a**) **HOF-a** and (**b**) **HOF-b**. **c** Simplified HOF structures with **hnb** topology in an AB-staggered fashion. Atom colors: C, gray; N, blue; H, white.

Supplementary Table 1). The space group of **MOF-b-Cu** determined by the PXRD pattern is $P3$, and unit-cell parameters are $a = 23.8643$ Å, $c = 6.9033$ Å, $V = 3404.8$ Å$^3$. The $a$-axis and the unit-cell volume of **MOF-b-Cu** are each 8.5% and 17.6% larger than those of **HOF-b**. On the other hand, the $c$-axis is almost unchanged. Scanning electron microscopy-energy dispersive spectroscopy (SEM-EDS) showed a uniform distribution of Cu in **MOF-b-Cu** (Supplementary Fig. 15). Broad bands at 3300 cm$^{-1}$ to 2800 cm$^{-1}$ in Fourier-transform infrared spectroscopy (FT-IR) of **HOF-b** suggest pyrazole N−H stretching. The bands disappeared in **MOF-b-Cu** (Supplementary Fig. 16). Thermogravimetric (TG) measurement in air determined the content of Cu to be 22.1 wt% in **MOF-b-Cu**. We calculated the ratio of Cu$^+$:**L2**$^{3-}$ is 3:1 (Supplementary Fig. 17 and Supplementary Table 2). This supports the quantitative metalation from [N−H⋯N] hydrogen bonds in **HOF-b** to [N−Cu−N] coordination bonds in **MOF-b-Cu**. We also prepared 60 μm crystals and conducted the metalation under the same conditions. 100% conversion with single crystallinity was observed (Supplementary Fig. 18).

The structure of **MOF-b-Cu** was optimized based on the unit cell refined by PXRD (Supplementary Fig. 9)[43–47]. **MOF-b-Cu** should have the same AB stacking model as **HOF-b**. This differs from the AA-stacking structure of a reported MOF prepared by ultrasonic reaction of 2,4,6-tris(4-aminophenyl)−1,3,5-triazine, 3,5-dimethyl-1H-pyrazole-4-carbaldehyde, and Cu$_2$O in 4 M aqueous acetic acid within 1 hour[48]. The three pyrazole rings in **MOF-b-Cu** are in coplanar configurations (Supplementary Fig. 13). The cell expansion originates from the longer N⋯N distance (3.518 Å) of [N−Cu−N] than that of [N−H⋯N] in **HOF-b** (2.509 Å, Fig. 2a and Supplementary Fig. 19), and enlarging the pore diameter to 1.9 nm. Solid-state $^{13}$C nuclear magnetic resonance (NMR, Supplementary Fig. 20) showed the carbon signals of two methyl groups of pyrazoles at 8 ppm and 12 ppm in **HOF-b** shifted to ~11 ppm in **MOF-b-Cu**. This supports the structural planarization. Scanning electron microscopy (SEM) showed that both **HOF-b** and **MOF-b-Cu** have hexagonal prism-like crystals (Fig. 2b, c, and Supplementary Fig. 21). The hexagonal channels along the $c$-axis of **MOF-b-Cu** were observed by high-resolution transmission electron microscopy (HR-TEM, Fig. 2d, e, and Supplementary Fig. 22). The interspace of 3.2 Å between lattice fringes matched the distance between adjacent structural layers. **MOF-b-Cu** showed a Type-I N$_2$ sorption profile at 77 K with a BET surface area of 2900 m$^2$ g$^{-1}$ and a pore size of 1.9 nm (Supplementary Fig. 23).

We tried synthesizing **MOF-b-Cu** by direct reactions of **H$_3$L2** with Cu(CF$_3$SO$_3$) under seven conditions (Supplementary Table 1). PXRD patterns revealed that all reactions yielded amorphous or low-crystalline products (Supplementary Fig. 24). X-ray absorption fine structure analysis confirmed that the amorphous samples have similar patterns to **MOF-b-Cu** (Supplementary Figs. 25–27), indicating that they have identical coordination environments around the Cu$^+$ ions to that of **MOF-b-Cu**. This result ruled out the possibility that the formation of **MOF-b-Cu** occurred by a dissolution-recrystallization process. Thermogravimetric analyses (TGA) showed that **MOF-b-Cu** does not show weight loss before 500 °C under Ar. The variable temperature PXRD patterns of **MOF-b-Cu** exhibited no change until 450 °C under Ar, indicating high thermal stability (Supplementary Figs. 28 and 29). We soaked **MOF-b-Cu** in boiling water, hot N,N-dimethylformamide (DMF), 1 M NaOH, and 3 M HCl aqueous solutions for two days, respectively. PXRD confirmed that their crystal structures are intact under the above-soaking conditions (Supplementary Fig. 30). The AB stacking structure of **MOF-b-Cu** was intact even after heating at 563 K, or metalation at 338 K, confirmed by differential scanning calorimetry and PXRD (Supplementary Figs. 31 and 32). The metalation was also applied to **HOF-a** and other metal ions (Supplementary Table 1). Under the same conditions to prepare **MOF-b-Cu**, we confirmed that 90.1%, 94.2%, and 93.2% of metalation occurred to have **MOF-a-Cu, MOF-b-Ag**, and **MOF-b-Au**, respectively. The analyses of

their structures and porosities are summarized in Supplementary Figs. 33–38 and Supplementary Table 2.

## Reaction mechanism of HOF-b to MOF-b-Cu

To understand the metalation reaction of **HOF-b**, we prepared seven samples in a methanol solution of 12 mM Cu(CF$_3$SO$_3$) at different soaking periods (0.5, 1, 2.5, 6, 10, 20, and 48 hours). Hereafter, **HOF | MOF-xh** denotes these samples, where $x$ refers to each soaking period. We carried out TGA in air to determine the Cu content (Fig. 2f, Supplementary Figs. 39 and 40). The ratio of metalation is 14.9% in **HOF | MOF-0.5 h** with the 1D axial heterostructure, and the time constant $t_{1/2}$, defined as the reaction time required to reach 50% metalation, was 7.4 h. Metalation at 48 mM and 2 mM Cu(CF$_3$SO$_3$) solutions was also carried out under the same conditions (Supplementary Fig. 41). These results show a consistent trend of metalation conversion from 0 to 10 h compared with that of the 12 mM solution. With the time increase up to 48 h, the degree of metalation in the 2 mM solution was only 82.7%. When we used Cu(BF$_4$) instead of Cu(CF$_3$SO$_3$), the metalation was 39.8% after soaking for 0.5 h, and the calculated $t_{1/2}$ is 1.0 h (Supplementary Table 3). This indicates that the BF$_4^-$ facilitates ion diffusion than CF$_3$SO$_3^-$, with coefficients of $1.07 \times 10^{-10}$ and $3.24 \times 10^{-11}$ cm$^2$ s$^{-1}$, respectively (Supplementary Fig. 42). Cu(CF$_3$SO$_3$) was used in the subsequent study of the metalation process, because of the slower kinetics which make it easier to stop the reaction at different stages (Supplementary Fig. 43)[49–52]. The [N−H⋯N] absorption band of FT-IR at 3300 to 2800 cm$^{-1}$ and 1250 cm$^{-1}$ weakens with an increasing soaking period, which is attributed to the metalation (Supplementary Fig. 44)[53].

We conducted Pair Distribution Function (PDF) analyses derived from synchrotron X-ray total scattering at 298 K to observe the evolution of Cu⋯Cu correlations upon metalation (Fig. 2g). **HOF | MOF-6h** shows three peaks at 3.1, 6.8, and 7.9 Å, all of which were not observed in **HOF-b**. The peak at 3.1 Å corresponds to Cu⋯Cu in the Cu$_3$ cluster, and the other two peaks correspond to the Cu⋯Cu from different layers (Supplementary Fig. 45). The intensity of the three peaks increases with the metalation period.

We conducted time-course ex-situ PXRD measurements. Figure 2h and Supplementary Figs. 46–48 showed that selected Bragg peaks related to the crystallographic $ab$ plane of **HOF-b** [100], [210], [220], [310], [300], and [420] shifted toward lower 2θ compared to that of **MOF-b-Cu**, respectively. The peak shifts indicate the structural expansion of the layer. As metalation occurs, the peak for [100] of the **HOF-b** domain at 4.64° gradually weakens. Meanwhile, the corresponding peak of the **MOF-b-Cu** at 4.35° increases. The emerging peak at 4.56° in **HOF | MOF-xh** ($x$ = 0.5, 1, 2.5, 6, 10, and 20) indicates the presence of an intermediate crystalline phase in the metalation. As a comparison, the peak of [002] remains at 26.22° during structural transformation, indicating that the interlayer spacing remains unchanged along the $c$-axis (Supplementary Fig. 47).

SEM-EDS provides the distribution of the Cu in **HOF | MOF-xh** ($x$ = 0, 2.5, 6, 20, and 48). The EDS line scan analysis of single crystals of **HOF | MOF-2.5 h** and **HOF | MOF-6h** along the $c$-axis showed that the concentration of Cu at the two end regions is higher than that of the central region (Fig. 3a, b, and Supplementary Figs. 49–52). Along the $ab$ plane, the line scans at the two ends and the central regions show a uniform distribution of Cu, further confirmed by the 2D EDS mapping (Fig. 3c, d). This indicates that the MOF | HOF | MOF crystals with the 1D axial heterostructure were formed. The Cu concentration in the central region increased in **HOF | MOF-20h**, and **MOF-b-Cu** has a uniform Cu distribution over the whole crystal, which indicates complete metalation. These results suggest that Cu$^+$ and CF$_3$SO$_3^-$ diffused the 1D channels of **HOF-b** along the $c$-axis, and the subsequent metalation takes place at the two end regions of the crystals.

BET surface areas of **HOF | MOF-xh** ($x$ = 0.5, 2.5, 10, 15, 20, and 48, Supplementary Fig. 53) determined by N$_2$ sorption isotherms at 77 K

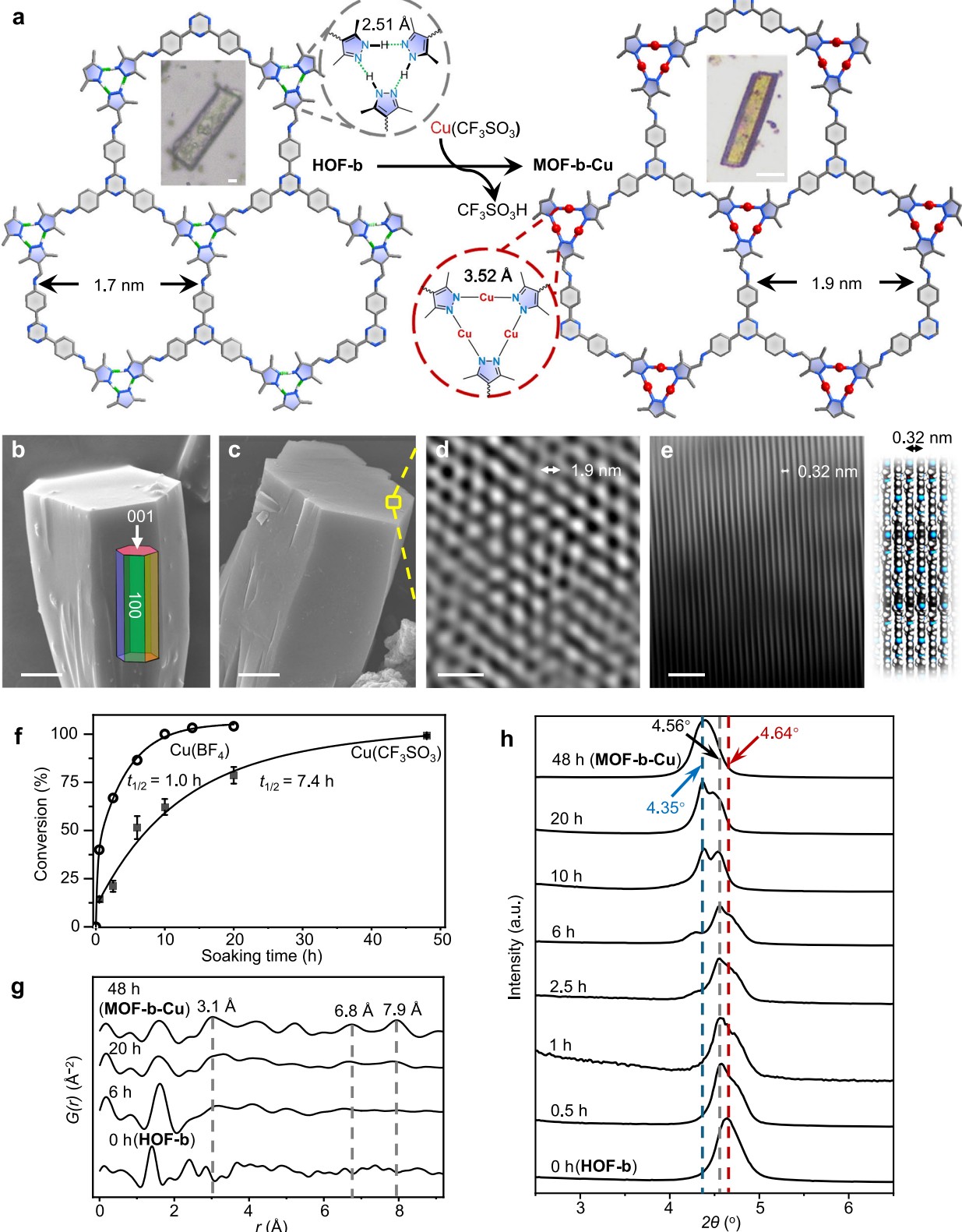

**Fig. 2 | Metalation of HOF-b. a** Crystal structures and optical images of **HOF-b** and **MOF-b-Cu**. Scale bar: 10 μm. Atom colors: C, gray; N, blue; H, white; Cu, red. SEM images of (**b**) **HOF-b** and (**c**) **MOF-b-Cu**. Scale bar: 5 μm. The HR-TEM images of (**d**) **MOF-b-Cu** on top view along the [002] direction (Scale bar: 5 nm) and (**e**) on side view along the relative *c*-axis as FFT pattern-filtered (Scale bar: 2 nm). Atom colors: C, dark gray; N, blue; H, white; Cu, red. **f** Metalation conversion during 48 h of $Cu(CF_3SO_3)$ and $Cu(BF_4)$. The mean and standard deviation of metalation conversion of $Cu(CF_3SO_3)$ across three metalation entries under the same conditions are indicated by the bars and error bars, respectively. **g** PDF profiles and (**h**) PXRD patterns of the metalation process with $Cu(CF_3SO_3)$.

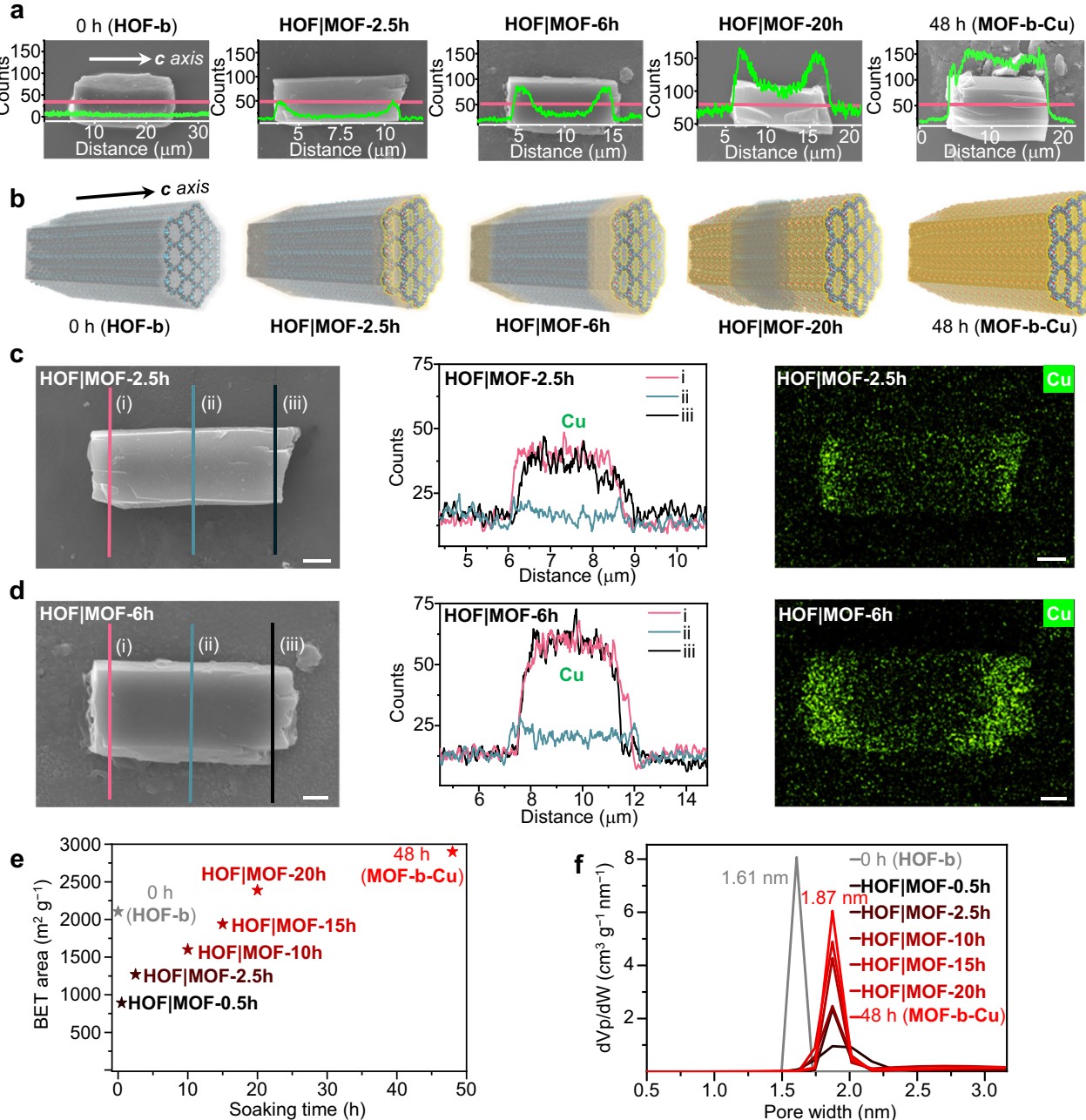

**Fig. 3 | Time-course line scan, mapping, and porosity of the HOF|MOF-*x*h with the 1D axial heterostructure. a** SEM-EDS line scan of single crystals for Cu element along the *c* axis of **HOF|MOF-*x*h** (*x* = 0, 2.5, 6, 20, and 48). **b** The proposed changes to the single crystals. Atom colors: C, gray; N, blue; H, white; Cu, red. SEM-EDS line scan and EDS mapping of (**c**) the **HOF|MOF-2.5 h** and (**d**) **HOF|MOF-6h** crystals along the *ab* plane. Scale bar: 1 μm. **e** BET surface areas of **HOF|MOF-*x*h** (*x* = 0.5, 2.5, 10, 15, 20, and 48), and (**f**) NLDFT calculated pore size distribution.

range from 893 (**HOF|MOF-0.5 h**) to 2388 (**HOF|MOF-20h**) m² g⁻¹, approaching the 2900 m² g⁻¹ of **MOF-b-Cu** (Fig. 3e and Supplementary Table 4). The NLDFT pore size distribution analysis shows that the **HOF|MOF-*x*h** (*x* = 0.5, 2.5, 10, 15, and 20) has the same pore size of 1.9 nm as that of **MOF-b-Cu** (Fig. 3f). With the increase in the metalation period, the peak of the pore size distribution increases. This indicates that $N_2$ molecules are mainly adsorbed in the MOF domains. The interface structure of MOF|HOF|MOF crystals may collapse after the solvent is removed, thus preventing the diffusion of gas molecules into the HOF domain[54,55].

## Photoluminescence

The solid-state UV-visible absorption spectrum (Supplementary Fig. 54) showed that **HOF|MOF-*x*h** (*x* = 0, 2.5, 6, and 48) have a deep

absorption band in the 365 to 425 nm region, which is attributed to the π-π* transition of the aromatic tri-pyrazole ligands[56]. **HOF-b** showed weak blue-green emission under 365 nm irradiation at 298 K, while **MOF-b-Cu, MOF-b-Ag**, and **MOF-b-Au** showed bright green, light yellow, and yellow emission, respectively (Fig. 4a, Supplementary Figs. 55 and 56). Under the excitation at 300 nm, **HOF-b, MOF-b-Cu, MOF-b-Ag**, and **MOF-b-Au** showed emission peaks at 465, 505, 519, and 545 nm, respectively (Fig. 4b and Supplementary Fig. 57). Both **HOF|MOF-2.5 h** and **HOF|MOF-6h** showed two emission peaks at 470 and 504 nm, consistent with those of **HOF-b** and **MOF-b-Cu** (Supplementary Fig. 58). We observed not only fluorescence bands depending on the composition ratios but also different emission bands from a single crystal (Fig. 4a and Supplementary Fig. 59). We measured quantum yields (QY) of luminescence at 298 K with fluorescence

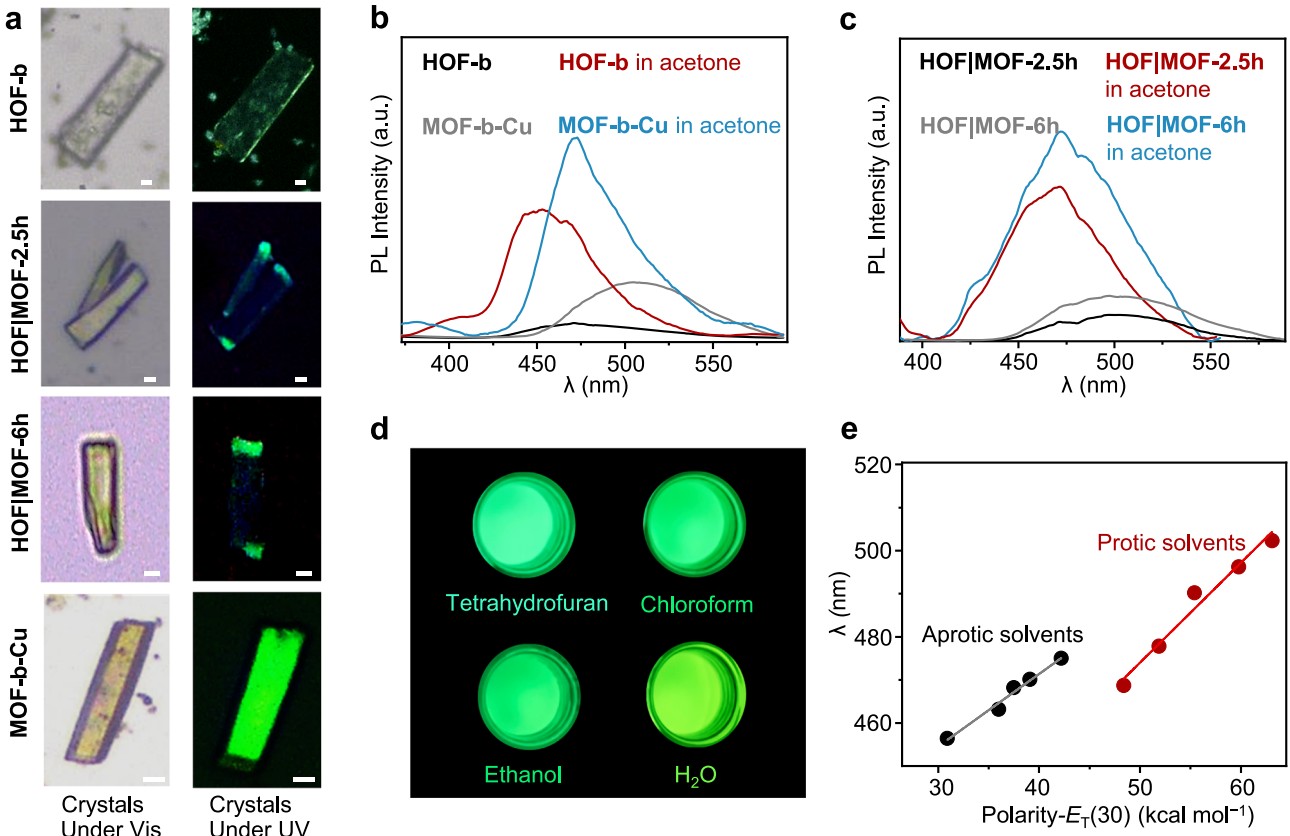

**Fig. 4 | Luminescence property. a** Optical images of **HOF│MOF-*x*h** (*x* = 0, 2.5, 6, and 48) crystals upon visible light (Vis) and UV excitation in the air at 365 nm. Scale bar: 5 μm. **b, c** The steady-state photoluminescence spectra (λ_ex = 300 nm). **d** Optical images of luminescence responses upon solvents of **MOF-b-Cu** under 365 nm irradiation. **e** Maximum emission against the solvent $E_T$(30) values of **MOF-b-Cu** (λ_ex = 300 nm).

standards. The QY of **HOF-b** and **MOF-b-Cu** are 2.8% and 4.9%, respectively. Their average lifetimes (0.83 ns for **HOF-b** and 1.14 ns for **MOF-b-Cu**) suggest the fluorescence (Supplementary Fig. 60). Previous studies of Cu₃, Ag₃, and Au₃-based complexes support the fact that the fluorescence mechanism of **MOF-b-Cu** is metal-to-ligand charge transfer (MLCT)[57,58]. The fluorescence properties of **MOF-b-Ag** and **MOF-b-Au** are summarized in Supplementary Table 5.

We investigated the fluorescence response behavior upon guest molecules. By soaking in acetone, **HOF-b** and **MOF-b-Cu** exhibited stronger emission peaks with blue shifts to 446 and 475 nm, respectively, compared to their guest-free states (Fig. 4b). **HOF│MOF-2.5 h** and **HOF│MOF-6h** in acetone showed emission peaks at 469 and 474 nm, respectively, which are between those of **HOF-b** and **MOF-b-Cu** (Fig. 4c). **MOF-b-Cu** in different solvents showed emission colors ranging from blue-green to yellow-green at 298 K (Fig. 4d). **MOF-b-Cu** demonstrated positive solvatochromic fluorescence in protic and aprotic solvents with a red shift in the emission band as solvent polarity increased (Fig. 4e). The emission peaks correlated with the empirical $E_T$(30)[59] parameter of Reichardt's dye B30 of solvent polarity values for ten solvents from 455 nm to 503 nm (Fig. 4e and Supplementary Table 6). The linear fits of the emission peaks against the $E_T$(30) values are obtained by separately plotting aprotic (slope = 1.68, R = 0.992) and protic solvents (slope = 2.32, R = 0.986). The difference in luminescent behaviors of **MOF-b-Cu** in polar protic and aprotic solvents arises from their hydrogen-bond donor ability toward the N atom of triazine groups[60]. Methanol would have a stronger hydrogen bonding interaction with **MOF-b-Cu** than other protic solvents, causing it to slightly deviate from the linear fit of the emission peaks against the polarity[61,62]. **MOF-b-Cu** is a visual indicator to distinguish various solvents.

We developed a method for constructing a 1D axial heterostructure of HOF and MOF crystals using the metalation process. The temporally and spatially controlled metalation reaction produced variable lengths of both HOF and MOF crystal domains in the 1D axial heterostructures. It resulted in controlling porosity and photoluminescence attributed to the size and positioning of the MOF crystals in the heterostructures, which is difficult to attain using single-phase crystals. The results would provide synthetic guidelines for creating new heterostructure crystals of porous molecular frameworks, even though each has distinct chemical bond characteristics. One of the limitations is that the proposed process to create 1D heterostructures only allows for metalation with monovalent metal ions because of the charge neutrality. Considering that the reported HOFs have a variety of accessible hydrogen-bonding sites and pore structures[39–41], this strategy is expected to apply to other types of HOFs with different metal ions and anions. Future efforts toward achieving channel continuity between these domains may unlock functionalities such as sequential gas capture and storage, multistep gas sieving. Moreover, expanding the system to multi-metallic 1D architectures (e.g., Ag-MOF│Cu-MOF │ HOF│Cu-MOF│Ag-MOF) could enable spatially controlled optical responses. These structurally engineered crystals are promising platforms for advanced applications in information storage and anticounterfeiting devices, as well as tunable axial charge carrier transport[29].

## Methods
### Synthesis of HOF-a
1 mmol of 1,3,5-benzenetricarboxaldehyde was added into a round-bottomed flask, and then 15 mL of ethanol and 200 μL of acetic acid were added into the system. 3.2 mmol of 3,5-Dimethyl-1H-pyrazol-4-

amine in 15 mL of ethanol was then added to the above solution. After reaction at 298 K for 24 hours, the product was filtered, and washed three times each with ethanol and methanol in succession. After degassing at 333 K for 6 hours, colorless crystals of **HOF-a** were obtained (400 mg, 91% yield). $^1$H NMR (594 MHz, DMSO-$d_6$) δ 12.31 (s, 3H), 8.68 (s, 3H), 8.36 (s, 3H), 2.30 (d, $J$ = 15.5 Hz, 18H). $^{13}$C NMR (149 MHz, DMSO-$d_6$) δ 154.55, 141.55, 138.69, 133.40, 128.36, 128.27, 13.72, 10.21.

### Synthesis of HOF-b

2,4,6-tris(4-aminophenyl)-1,3,5-triazine (1 mmol) and 3,5-dimethyl-1H-pyrazole-4-carbaldehyde (3.2 mmol) were placed in a round-bottomed flask (250 mL). Then 30 mL of ethanol and 200 μL of acetic acid were added to the above flask. After stirring at 363 K for 30 min, 10 mL of *N,N*-dimethylformamide was added to the reaction system and refluxed for 24 hours. The mixture was then cooled to 298 K, and 150 mL of methanol was added to the flask to allow **HOF-b** to recrystallize from the solvent at 253 K overnight. The mixture was filtered under reduced pressure, and the product was washed three times each with ethanol and methanol in succession. After degassing at 333 K for 6 hours, colorless crystals (560 mg, 83% yield) of **HOF-b** were obtained. $^1$H NMR (594 MHz, DMSO-$d_6$) δ 12.71 (s, 3H), 8.71 (d, $J$ = 8.8 Hz, 6H), 8.52 (s, 3H), 7.36 (d, $J$ = 7.9 Hz, 6H), 2.38 (s, 18H). $^{13}$C NMR (149 MHz, DMSO-$d_6$) δ 170.47, 157.35, 155.37, 149.09, 142.75, 131.78, 129.86, 121.45, 113.93, 13.42, 10.27.

### Metalation of HOF-a and HOF-b

0.1 mmol of **HOF-b** and 25 mL of dry methanol were added to a glass vial (100 mL) in the glove box under an Ar atmosphere. Then, 0.6 mmol of Cu(CF$_3$SO$_3$), Ag(CF$_3$SO$_3$), or Au(CF$_3$SO$_3$) (synthesized by AuCl and Na(CF$_3$SO$_3$) in dry methanol) in 25 mL of dry methanol (12 mM) were added to the above solution, respectively. Metalation was carried out at 298 K for 48 hours. The products were exchanged in dry methanol three times in 2 days. Finally, the products were filtered and washed three times with dry DMF and methanol in succession. After degassing at 333 K for 12 hours, orange-yellow crystals of **MOF-b-Cu** (79.6 mg, 92% yield), bright yellow crystals of **MOF-b-Ag** (83.9 mg, 85% yield), and golden yellow crystals of **MOF-b-Au** (103.2 mg, 82% yield) were obtained. **MOF-b-Cu-338K** was synthesized from **HOF-b** at 338 K for 48 hours under the same conditions (84.6 mg, 98% yield).

**HOF|MOF-xh** ($x$ = 0.5, 1, 2.5, 6, 10, and 20) were prepared by soaking **HOF-b** crystals in a dry methanol solution of Cu(CF$_3$SO$_3$) at the corresponding periods. The products were exchanged in the dry methanol three times in 2 days. Finally, the products were filtered and washed three times with dry methanol. After degassing at 298 K for 12 hours to obtain the crystals of **HOF|MOF-0.5 h** (66.7 mg, 77% yield), **HOF|MOF-1h** (69.2 mg, 80% yield), **HOF|MOF-2.5 h** (70.5 mg, 82% yield), **HOF|MOF-6h** (74.0 mg, 86% yield), **HOF|MOF-10h** (73.8 mg, 86% yield), and **HOF|MOF-20h** (78.9 mg, 92% yield), respectively. **MOF-a-Cu** was synthesized from **HOF-a** for 48 hours using Cu(BF$_4$) (synthesized by Cu$_2$O and HBF$_4$ in dry methanol) (57.4 mg, 91% yield). **HOF|MOF-xh-BF$_4$** ($x$ = 0.5, 2.5, 6, 10, 14, and 20) were prepared by soaking **HOF-b** crystals into a methanol solution of Cu(BF$_4$) at the corresponding periods. The crystals of **HOF|MOF-1h-BF$_4$** (73.5 mg, 85% yield), **HOF|MOF-2.5h-BF$_4$** (75 mg, 87% yield), **HOF|MOF-6h-BF$_4$** (79.8 mg, 93% yield), **HOF|MOF-10h-BF$_4$** (78.6 mg, 91% yield), **HOF|MOF-14h-BF$_4$** (81.0 mg, 94% yield), and **HOF|MOF-20h-BF$_4$** (80.4 mg, 93% yield), respectively.

### Metalation of HOF-b under other concentrations

**2 mM solution:** 0.1 mmol of **HOF-b** and 150 mL of dry methanol were added to a glass vial (500 mL) in the glove box under an Ar atmosphere. Then 0.6 mmol of Cu(CF$_3$SO$_3$) in 150 mL of dry methanol (2 mM) were added to the above solution, respectively. Metalation was carried out at 298 K for 48 hours. The products were exchanged in dry

methanol three times in 2 days. Finally, the products were filtered and washed three times with dry DMF and methanol in succession. After degassing at 333 K for 12 hours, orange-yellow crystals of **MOF-b-Cu-2mM** (60.8 mg, 70% yield).

**48 mM solution:** 0.1 mmol of **HOF-b** and 6.25 mL of dry methanol were added to a glass vial (100 mL) in the glove box under an Ar atmosphere. Then 0.6 mmol of Cu(CF$_3$SO$_3$) in 6.25 mL of dry methanol (48 mM) was added to the above solution, respectively. Metalation was carried out at 298 K for 48 hours. The products were exchanged in the dry methanol three times in 2 days. Finally, the products were filtered and washed three times with dry DMF and methanol in succession. After degassing at 333 K for 12 hours, orange-yellow crystals of **MOF-b-Cu-48mM** (78.5 mg, 91% yield).

### Direct synthesis of MOF-b-Cu

0.1 mmol of **HOF-b** was dissolved in 25 mL of dry DMF in a glass vial in the glove box. Then 0.6 mmol of Cu(CF$_3$SO$_3$) in 25 mL of dry DMF was added to the above solution. The reaction was carried out at 298 K for 48 hours. The products were exchanged in dry methanol three times in 2 days. Finally, the product was filtered and washed three times with dry DMF and methanol in succession. After degassing at 60 °C for 12 hours, green-yellow powders were obtained: 57.4 mg (67 % yield).

### General measurements

Solution NMR spectra were measured on a Bruker Avance III instrument with an AS500 magnet at 298 K. FT-IR spectra were measured on a Bruker Optics ALPHA FT-IR spectrometer with a Universal ATR accessory in the range of 4000–500 cm$^{-1}$ at 298 K. N$_2$ sorption measurements were measured on a BELSORP MINI-X and BELSORP MAX (BEL-Japan, Inc.) at 77 K. The samples were activated at 353 K under vacuum for 12 hours. UV-vis absorbance spectroscopy was measured with a Shimadzu UV-3600 at 298 K from 800 to 200 nm on the sample/BaSO$_4$ mixture plate at 298 K. TGA was measured using a Rigaku TG-DTA8121 under flowing Ar and a Rigaku TG-DTA8122 under flowing air. The TGA profiles in Ar were obtained by heating from -298 K to 773 K at a rate of 10 K min$^{-1}$. The TGA profiles in air were obtained by using two different heating programs at a rate of 10 K min$^{-1}$. **HOF-a, HOF-b, MOF-b-Cu**, and **MOF-b-Au** were measured from -298 K to 1173 K. **MOF-a-Cu, MOF-b-Ag, HOF|MOF-xh** ($x$ = 0.5, 1, 2.5, 6, 10, and 20), and **HOF|MOF-xh-BF$_4$** ($x$ = 0.5, 2.5, 6, 10, 14, and 20) were measured from -298 K to 773 K and then held for 30 min. A differential scanning calorimetry experiment was performed on a Hitachi model DSC7020, High-Tech Science Corporation, under a N$_2$ atmosphere with a heating rate of 10 K min$^{-1}$. Optical images under visible light were captured using a Leica DM2700 M bright-field microscope at 298 K. Photoluminescence images under UV were captured on an Olympus BX53MTRF-S optical microscope at 298 K.

### X-ray measurements

The SCXRD measurement was tested on a Rigaku AFC10 diffractometer with a Rigaku Pilatus P200K system equipped with a MicroMax-007 HF/Varimax rotating-anode X-ray generator with confocal monochromated MoKα radiation. Then, the single-crystal structure was resolved and refined using direct methods with Olex2. PXRD patterns were measured on a Rigaku MiniFlex diffractometer with CuKα anode at 298 K. Variable-temperature PXRD measurements were performed with a BTS 500 benchtop heating stage (Anton Paar GmbH Company, Austria) at temperatures ranging from 303 to 723 K at a heating rate of 10 K min$^{-1}$ and held at each target temperature for 10 min under flowing Ar.

### Simulation of crystal structures[63,64]

The structure models of **HOF-b** and **MOF-a-Cu** were created from **HOF-a** and optimized based on the unit cell refined by their PXRD patterns. The structure models of **MOF-b-Cu, MOF-b-Ag**, and **MOF-b-**

**Au** were created from **HOF-b** and optimized based on the unit cell refined by their PXRD patterns. The Forcite module of Material Studio was used to optimize the model structures. Using Universal Force Field (UFF) parameters and Ultra-fine quality mode, only internal coordinates were optimized first. Both internal coordinates and cell parameters were optimized. The unit cell with the highest symmetry was selected for the structure. It was also recommended to use CASTEP or DMol3 for the geometry optimization of the unit cell. The output CIF file was used for PXRD pattern simulation. Rietveld analysis for their structures was simulated by FullProf Suite (FPS) ToolBar software based on experimental PXRD patterns.

### X-ray absorption fine structure measurements[65]
The spectra at the Cu K-edge were acquired on the BL14B2 beamline at Super Photon ring-8 GeV (SPring-8, Hyogo, Japan). All samples were activated at 318 K under vacuum for 12 hours before measurement. All samples were mixed with boron nitrile and pressed into pellets (10 mm diameter, ~0.5 mm thickness) and sealed in an Ar-filled glovebox. The collected data at 298 K were then extracted and processed following the standard procedures using the ATHENA module with IFEFFIT software packages. The $k^3$-weighted X-ray absorption fine structure (EXAFS) spectra were obtained by subtracting the postedge background from the corresponding total absorption and then normalized regarding the edge-jump step. The $k^3$-weighted $\chi(k)$ data were then Fourier transformed into the real ($R$) space by using Hanning windows ($d_k = 1.0$ Å$^{-1}$) to separate these EXAFS contributions from the different coordination shells. Fourier transformation was $k^3$-weighted in the $k$ range of 2.8–10.4 Å$^{-1}$. All the data were without the phase shift correction.

### X-ray total scattering measurements
All samples were sealed in borosilicate capillaries ($\phi = 0.7$ mm) in an Ar-filled glove box. The X-ray total scattering data were collected at 298 K with two 2D CdTe detectors at the BL04B2 beamline at SPring-8. The incident energy was 112.9232 keV. $G(r)$ was obtained from the Fourier transform of $S(Q)$ with a Lorch modification function by using Igor Pro software[66].

### Electron microscopy and EDS measurements
SEM, EDS mapping, and EDS line scan images were measured using a JEOL JSM-IT500HR at 298 K. HR-TEM was measured on a JEOL JEM-2200FS equipped with a field emission gun under an operrating voltage of 200 kV at 298 K. HR-TEM and selected area electron diffraction (SAED) were performed to probe the crystallinity. Due to the sensitivity of MOFs to electron irradiation, low-dose electron microscopy was performed.

### Photoluminescence measurements and quantum yield calculation[67,68]
Photoluminescence spectra were recorded using a JASCO FP-6600DS spectrofluorometer at 298 K. Photoluminescence lifetime measurements were measured by Quantaurus-Tau-C11367, Hamamatsu Photonics. The fluorescence QY was calculated using Eq. (1) with an ethanol solution of rhodamine 6 G ($\Phi_R = 0.95$, 298 K) as a reference under identical instrument settings/measurement conditions. The R in the subscript indicates the relevant parameter of rhodamine 6 G; $\Phi$ is the fluorescence QY (%) of samples; $I$ is the integral fluorescence ($\lambda_{em}$); $A$ is the corresponding absorption factor ($\lambda_{ex}$), and $n$ is the refractive index of the solvent.

$$\phi = \phi_R \left(\frac{I}{I_R}\right)\left(\frac{A_R}{A}\right)\left(\frac{n}{n_R}\right)^2 \qquad (1)$$

### Diffusion coefficient calculation
The conversion from HOF to MOF has two processes: (i) diffusion of ions into the pores (ii) exchange from H$^+$ to Cu$^+$. The fact that anion size affects diffusion rate suggests that diffusion is the rate-limiting step in the conversion. We then estimated diffusion coefficients from the conversion rate profiles by fitting to Fick's second law of diffusion Eq. (2), which was widely applied to discuss the diffusion behaviors in various MOF crystals[69–71].

$$\frac{M_t}{M_\infty} = 1 - \frac{8}{\pi^2}\sum_{n=0}^{\infty}\frac{1}{(2n+1)^2}\exp\left(-\frac{D(2n+1)^2\pi^2 t}{4L^2}\right) \qquad (2)$$

Here, $t$ is the soaking time. $M_t$ is the amount of Cu$^+$ incorporated into the crystal at time $t$. $M_\infty$ represents the Cu$^+$ loading at saturation (48 hours). $D$ is the diffusion coefficient. $L$ is the effective diffusion length. The average length of **HOF-b** single crystals is approximately 35 μm (Supplementary Fig. 18). Since the diffusion occurs from both ends of the crystal, $L$ is taken as 17.5 μm. $n$ is the number of series in the summation. Since the ions diffuse along the 1D channels in the crystal in this work, the first-order approximation (n = 0) of the solution to Fick's second law Eq. (3) was employed for analysis.

$$\ln\left(1 - \frac{M_t}{M_\infty}\right) = -\frac{\pi^2 D}{4L^2}t + \ln\left(\frac{8}{\pi^2}\right) \qquad (3)$$

To improve the accuracy of the fitting, the diffusion coefficient is determined using data from the initial stage (0–10 h) of the diffusion process (Fig. 2f).

## Data availability
Data supporting the findings of the study are available in the paper and its Supplementary Information. Source data are provided with this paper. Crystallographic data for the structure reported in this Article have been deposited at the Cambridge Crystallographic Data Centre, under deposition no. CCDC 2433227 (**HOF-a**). Copies of the data can be obtained free of charge at https://www.ccdc.cam.ac.uk/structures/. Source data are provided with this paper.

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

## Acknowledgements

S.Z. acknowledges the China Scholarship Council (CSC, 1060-34-3142). The work was supported by the Japan Society of the Promotion of Science (JSPS) for a Grant-in-Aid for Scientific Research (B) (JP18H02032), Challenging Research (Exploratory) (JP19K22200), Transformative Research Areas (A) "Supra-ceramics" (JP22H05147), Fund for the Promotion of Joint International Research (International Collaborative Research, JP24K0112). Y.-S.W. acknowledges the Murata Science and Education Foundation, Research Institute for Production Development, The Foundation for the Promotion of Ion Engineering, and Iketani Science and Technology Foundation. L.L. acknowledges the JSPS Postdoctoral Fellowships for Research in Japan (P21761). C.L. acknowledges the CSC and the Ministry of Education, Culture, Sports, Science and Technology (MEXT) of Japan. The authors thank Ms. Nanae Shimanaka for the SCXRD measurements in the Institute for Integrated Cell-Material Sciences, Kyoto University.

## Author contributions

S.H., Y.-S.W., and S.Z. designed this work. S.H., Y.-S.W., and S.Z. wrote this manuscript. S.Z. did organic synthesis, characterization tests, and simulations. L.L. helps with the HOF crystal synthesis. E.K.B. helped in SCXRD and SEM-EDS. M.T. helped with the HR-TEM measurements. Z.F. helps with the UV-vis and fluorescence spectra. C.L. helped with the TGA measurements. Y.-S.W. helped with $N_2$ sorption measurements. N.H. helped with the solid-state NMR data. All authors have approved the final version of the manuscript.

## Competing interests

The authors declare no competing interests.
