## [Transparent Peer Review file · Nature Communications]

1D axial heterostructure of hydrogen-bonded framework and metal-organic framework by metalation reaction

Corresponding Author: Professor Satoshi Horike

Version 0:

Reviewer comments:

Reviewer #1

(Remarks to the Author)

In the manuscript, a method for the preparation of 1D axial heterostructures by metallization strategies was developed. In this study, rod-shaped single crystals of HOFs with [N–H···N] hydrogen bonds were immersed in Cu solution to induce metallization, resulting in tunable crystals with 1D axial heterostructures (MOF|HOF|MOF). The authors have done a lot of research on the structure of the synthesized material. However, the significance of this study and its practical application value are not detailed in the manuscript. In addition, the performance advantages of the MOF|HOF|MOF have not been studied. Therefore, the current manuscript requires a significant revision. Here are some suggestions for further revision.

1. What are the advantages of 1D axial heterostructures over other heterostructures?
2. What are the current methods for preparing one-dimensional heterostructures? What are the significant advantages of this study's methodology?
3. What performance of MOF|HOF|MOF has been improved, and what is its value in practical applications? Experimentally proven.
4. The kinetics of Cu⁺ diffusion along the 1D channel (e.g., activation energy calculations or diffusion coefficient estimations) need to be explored in depth to support the conclusion that anion size affects diffusion rate (Figure 2F)
5. What is the trend of the influence of Cu⁺ solution concentration on the metallization process of HOF?
6. Failure of direct synthesis of MOF-b-Cu results in amorphous products (lines 130-132), and it is recommended to analyze the amorphous structure by XRD or EXAFS to verify its differences from metallized products.
7. The key innovations of present work can be emphasized in the last part of introduction section. The shortcomings of present work/method can be emphasized in the conclusion section.
8. Please provide the approximate temperature of "room temperature". The unit "minutes" can be changed to "min".
9. Please define the abbreviations at they are first used in the text, as there are many abbreviations have been defined in the "Methods" section.

Reviewer #2

(Remarks to the Author)

The authors, Wei and Horike et al., constructed 1D axial heterostructures composed of HOF and MOF parts by immersing HOF crystals into a Cu cation solution. The hydrogen-bonded trimer of pyrazole nicely converted into the isostructural SBU. Crystal structures of both the mother HOFs and generated MOFs are well characterized by X-ray diffraction, HR-TEM, PDF and SEM-EDS. Particularly, SEM-EDS clearly shows the heterostructure of the crystals.

Surface area and photo fluorescence properties obviously depend on the ratio of HOF and MOF parts. The present manuscript shows a very elegant proof-of-concept MOF/HOF/MOF heterostructures, and therefore, can open a door for a new class of molecular-based porous materials. The manuscript can be suitable for publication in Nature Communications after considering the following comments:

1. Have you obtained time course optical macroscopic images of the same single crystals? The change of the crystalline morphology, particularly the width of crystals, should be changed. Regarding this, is there size limitation of the crystals for perfect metalation with keeping single crystallinity.
2. In Figure 4E, the slope lines are not straight but angulated.

Reviewer #3

(Remarks to the Author)

The manuscript from Wei and Horike reported the transformation of a hydrogen-bonded organic framework (HOF) to a metal organic framework (MOF) through metal metallization. The metalation time control study further revealed that a 1D axial MOF|HOF|MOF heterostructure can be achieved. The crystals exhibited interface-controlled gas diffusion and various spatially-resolved photoluminescence behaviors depending on the distribution of each component. Overall, the concept of MOF|HOF heterostructure is remarkable, and the conclusions are supported by a series of thorough characterizations. I suggest publish this work in Nat. Commun. after the following points are addressed.

(1) Recently, Dan Li et al. reported JNU-304 in J. Am. Chem. Soc. 2025, 147, 16, 13711–13720. It has the same layer structure with MOF-b-Cu, although the packing looks different. The authors need to mention this work and discuss the difference.

(2) Strong interaction between the $\text{Cu}_3(\text{pyrazolate})_3$ species are well known, so is there any possibility for MOF-b-Cu to have different stacking configuration with the HOF-b?

Version 1:

Reviewer comments:

Reviewer #1

(Remarks to the Author)

In the manuscript, a method for the preparation of 1D axial heterostructures by metallization strategies was developed. After the author's revision, the manuscript was further refined and supplemented. However, the current manuscript requires minor revisions. Here give some suggestions for further revision.

1. The symbol for the range between numbers is "-" instead of "-"; The font of the formula should be consistent with the body (Page 8, Line 452).
2. In the subsection "Metalation of HOF-a and HOF-b", the author mainly uses MOF-b-Cu as the object of analysis. So, what is the reason for choosing MOF-b-Cu over MOF-a-Cu, MOF-b-Ag, or MOF-b-Au? What are the advantages of MOF-b-Cu over them?
3. Does this metallization method work with other types of HOF materials?

Reviewer #2

(Remarks to the Author)

The manuscript has been well improved according to the reviewers' comments, and is acceptable for Nature Communications.

Reviewer #3

(Remarks to the Author)

In this revised manuscript, the authors have successfully addressed all the questions by the referees, I recommend this manuscript for publication.

RESPONSE TO REVIEWERS' COMMENTS

Responses to Reviewer #1:

In the manuscript, a method for the preparation of 1D axial heterostructures by metallization strategies was developed. In this study, rod-shaped single crystals of HOFs with [N–H···N] hydrogen bonds were immersed in Cu solution to induce metallization, resulting in tunable crystals with 1D axial heterostructures (MOF|HOF|MOF). The authors have done a lot of research on the structure of the synthesized material. However, the significance of this study and its practical application value are not detailed in the manuscript. In addition, the performance advantages of the MOF|HOF|MOF have not been studied. Therefore, the current manuscript requires a significant revision. Here are some suggestions for further revision.

1. What are the advantages of 1D axial heterostructures over other heterostructures?

Re: We can find other heterostructures such as core–shell, superlattice, graded, and Janus structures. The unique features and advantages of the 1D axial heterostructures, among others, are mainly based on their highly anisotropic (1D) crystal arrays. These structural features endow them with properties, including multiple spatially resolved emissions and tunable axial charge carrier transport.

We revised the “Introduction” on Page 2, paragraph 2: “The unique features and advantages of the 1D axial heterostructures compared with other heterostructures (core-shell, superlattice, graded, and Janus structures) are mainly based on their highly anisotropic (1D) crystal arrays.”

2. What are the current methods for preparing one-dimensional heterostructures? What are the significant advantages of this study's methodology?

Re: The reported primary methods for preparing 1D heterostructures are sequential growth using parent crystals (Refs. 4 and 5). On the other hand, we applied the metalation method in this work. The advantages of this approach are that it can provide precise control over the size and distribution of HOF and MOF domains by overcoming different chemical bonds and controlling the metalation process.

We added these points in the revised manuscript (Page 2, paragraph 2; Page 3, paragraph 2): “which is usually produced by sequential growth using parent crystals.^{4,5}”

“This approach provides precise control over the size and distribution of HOF and MOF domains by overcoming different chemical bonds and controlling the metalation process.”

3. What performance of MOF|HOF|MOF has been improved, and what is its value in practical applications? Experimentally proven.

Re: Regarding the materials functionality in this work, we conducted (i) control of the BET surface areas in the 1D heterostructures depending on the ratio of HOF and MOF (Table S4), (ii) control of 1D anisotropic spatially-resolved luminescence (Figures 4 and S59) depending on the size and distribution of HOF and MOF in the 1D heterostructures. These are uniquely observed in the 1D heterostructures and do not represent an improvement compared to the single phases of HOF or MOF in this work.

On the other hand, as the reviewer suggested, we have not presented the practical applications specifically attributed to the 1D heterostructures. One of the reasons is that 1D pore connectivity at the interface of HOF and MOF is not optimized because of the lattice mismatch (pore diameters of the HOF and MOF are 16 Å and 19 Å). Although we confirmed that the HOF and MOF crystal domains in the heterostructures are mutually connected, the gas sorption isotherm experiments (Figure S53) suggest that the N₂ molecules do not pass through the MOF to the HOF, because of pore dysconnectivity. If we could synthesize the HOF-MOF 1D heterostructure having channel connectivity, we could expect more functionalities such as sequential gas capture and storage, multistep gas sieving, and so on.

Another potential function is the multi-domain and multi-spatially resolved optical properties via HOF to MOF transformation. In the present work, we demonstrate the metalation of HOF using single metal ions (Cu⁺, Ag⁺, Cu⁺) to create the 1D heterostructure. These results can be extended to the creation of multi-metal-exchanged systems, such as Ag-MOF|Cu-MOF|HOF|Cu-MOF|Ag-MOF, in 1D heterostructures. These multi-component 1D heterostructural crystals are potentially applicable for information storage and anticounterfeiting devices, as well as tunable axial charge carrier transport.

In this work, we focused on the conceptualization of creating 1D heterostructures of chemically mismatched HOF and MOF using the metalation process, and believe the above ideas for applications are suitable for future work.

Accordingly, we revised the manuscript to clarify these points on Page 2, paragraph 2; Page 13, paragraph 2:

“This provides potential applications such as information storage and axial carrier transport for solar energy conversion.^{15, 16”}

“Future efforts toward achieving channel continuity between these domains may unlock functionalities such as sequential gas capture and storage, multistep gas sieving.

Moreover, expanding the system to multi-metallic 1D architectures (e.g., Ag-MOF|Cu-MOF|HOF|Cu-MOF|Ag-MOF) could enable spatially controlled optical responses. These structurally engineered crystals are promising platforms for advanced applications in information storage and anticounterfeiting devices, as well as tunable axial charge carrier transport.²⁹

4. The kinetics of Cu⁺ diffusion along the 1D channel (e.g., activation energy calculations or diffusion coefficient estimations) need to be explored in depth to support the conclusion that anion size affects diffusion rate (Figure 2F)

Re: The metalation-induced conversion from HOF to MOF has two processes: (i) diffusion of ions into the pores (ii) exchange from H⁺ to Cu⁺. The fact that anion size significantly affects diffusion rate suggests that diffusion is the rate-limiting step in the conversion. We then tried to estimate diffusion coefficients from the conversion rates (Figure S42). To calculate the diffusion coefficients, we applied the Fickian diffusion model, which was discussed and applied to various solids in reported works. For instance, organic solvents transport through UiO-66 (*J. Phys. Chem. C*, 2018, 122, 16060); water adsorption in MOF-303 (*Energy Convers. Manag.*: X, 2024, 24, 100694); 1,3,5,7-tetramethyl-4,4-difluoroboradiazaindacene diffusion in NU-1008 (*Langmuir* 2020, 36, 36, 10853); sulfate and chloride ions diffusion in concrete (*Sci. Rep.* 2025, 15, 12480); volatile organic vapors diffusion in organic films (*Mol. Syst. Des. Eng.*, 2020, 5, 1057). Based on the above cases, we think that Fick's second law is applicable to describe the diffusion behavior of ions in this work.

The calculation details and discussion were added to the revised main manuscript on Page 8, paragraph 2, and Page 19, paragraph 2: "This indicates that the BF₄⁻ facilitates ion diffusion than CF₃SO₃⁻, with coefficients of 1.07×10⁻¹⁰ and 3.24×10⁻¹¹ cm² s⁻¹, respectively (Figure S42)."

"The conversion from HOF to MOF has two processes: (i) diffusion of ions into the pores (ii) exchange from H⁺ to Cu⁺. The fact that anion size affects diffusion rate suggests that diffusion is the rate-limiting step in the conversion. We then estimated diffusion coefficients from the conversion rate profiles by fitting to Fick's second law of diffusion (Eq. 1), which was widely applied to discuss the diffusion behaviors in various MOF crystals.⁶⁹⁻⁷¹

$$\frac{M_t}{M_\infty} = 1 - \frac{8}{\pi^2} \sum_{n=0}^{\infty} \frac{1}{(2n+1)^2} \exp\left(-\frac{D(2n+1)^2 \pi^2 t}{L^2}\right) \quad (1)$$

Here, t is the soaking time. M_t is the amount of Cu^+ incorporated into the crystal at time t . M_∞ represents the Cu^+ loading at saturation (48 hours). D is the diffusion coefficient. L is the effective diffusion length. The average length of **HOF-b** single crystals is approximately 35 μm (Figure S18). Since the diffusion occurs from both ends of the crystal, L is taken as 17.5 μm . n is the number of series in the summation. Since the ions diffuse along the 1D channels in the crystal in this work, the first-order approximation ($n = 0$) of the solution to Fick's second law (Eq. 2) was employed for analysis.

$$\ln\left(1 - \frac{M_t}{M_\infty}\right) = -\frac{\pi^2 D}{4L^2} \cdot t + \ln\left(\frac{8}{\pi^2}\right) \quad (2)$$

To improve the accuracy of the fitting, the diffusion coefficient is determined using data from the initial stage (0~10 h) of the diffusion process (Figure 2F)."

5. What is the trend of the influence of Cu^+ solution concentration on the metallization process of HOF?

Re: We tried two other concentrations (48 mM and 2 mM) of $\text{Cu}(\text{CF}_3\text{SO}_3)$ for the metalation. The results show a consistent trend of metalation conversion from 0 to 10 h compared with that of the 12 mM solution (Figure S41). With the time increase up to 48 h, the degree of metalation in the 2 mM solution is only 82.7%.

The experiments and results were added to the revised main manuscript (Page 8, paragraph 2): "Metalation at 48 mM and 2 mM $\text{Cu}(\text{CF}_3\text{SO}_3)$ solutions was also carried out under the same conditions (Figure S41). These results show a consistent trend of metalation conversion from 0 to 10 h compared with that of the 12 mM solution. With the time increase up to 48 h, the degree of metalation in the 2 mM solution was only 82.7%."

6. Failure of direct synthesis of MOF-b-Cu results in amorphous products (lines 130-132), and it is recommended to analyze the amorphous structure by XRD or EXAFS to verify its differences from metallized products.

Re: We measured PXRD and synchrotron X-ray absorption (XAS) for the amorphous samples. PXRD patterns revealed that all reactions of direct synthesis yielded amorphous or low-crystalline products (Figure S24). XAS analysis confirmed that the amorphous samples have similar patterns to **MOF-b-Cu** (Figures S25-S27), indicating that they have identical coordination environments around the Cu^+ ions to that of **MOF-b-Cu**.

The results and explanation were added to the revised main manuscript (Page 7, paragraph 2): "PXRD patterns revealed that all reactions yielded amorphous or low-crystalline

products (Figure S24). X-ray absorption fine structure analysis confirmed that the amorphous samples have similar patterns to **MOF-b-Cu** (Figures S25-S27), indicating that they have identical coordination environments around the Cu^+ ions to that of **MOF-b-Cu**.”

7. The key innovations of present work can be emphasized in the last part of introduction section. The shortcomings of present work/method can be emphasized in the conclusion section.

Re: Traditionally, the sequential crystal growth process has been applied to create 1D heterostructures in various substances. It prepares the parent crystals and then grows the second crystal. However, this approach is limited to the same type of chemical bonds in two or more crystals. In this work, we used metalation-based post-synthetic approach. This offers the advantage of enabling precise control over the size and ratio of chemical bond-mismatched HOF and MOF domains in the heterostructure. On the other hand, this method has one shortcoming: the metalation process for HOFs is limited to monovalent metal ions at this moment, because of the charge neutrality. It is not possible to use divalent or multivalent metal ions.

We revised the parts of “Introduction” and “Conclusion” as follows: “This approach provides precise control over the size and distribution of HOF and MOF domains by overcoming different chemical bonds and controlling the metalation process.”

“One of the limitations is that the proposed process to create 1D heterostructures only allows for metalation with monovalent metal ions because of the charge neutrality.”

8. Please provide the approximate temperature of “room temperature”. The unit “minutes” can be changed to “min”.

Re: We changed the “room temperature” and “minutes” to “298 K” and “min”, respectively, in the revised manuscript and supporting information.

9. Please define the abbreviations at they are first used in the text, as there are many abbreviations have been defined in the “Methods” section.

Re: We confirmed that all the abbreviations were defined as they are first used in the revised manuscript.

Responses to Reviewer #2:

The authors, Wei and Horike et al., constructed 1D axial heterostructures composed of HOF and MOF parts by immersing HOF crystals into a Cu cation solution. The hydrogen-bonded trimer of pyrazole nicely converted into the isostructural SBU. Crystal structures of both the mother HOFs and generated MOFs are well characterized by X-ray diffraction, HR-TEM, PDF and SEM-EDS. Particularly, SEM-EDS clearly shows the heterostructure of the crystals.

Surface area and photo fluorescence properties obviously depend on the ratio of HOF and MOF parts. The present manuscript shows a very elegant proof-of-concept MOF/HOF/MOF heterostructures, and therefore, can open a door for a new class of molecular-based porous materials. The manuscript can be suitable for publication in Nature Communications after considering the following comments:

1. Have you obtained time course optical macroscopic images of the same single crystals? The change of the crystalline morphology, particularly the width of crystals, should be changed. Regarding this, is there size limitation of the crystals for perfect metalation with keeping single crystallinity.

Re: We performed *in situ* optical microscopy observations on two single crystals during the metalation process (Figure R1 below). Throughout the process, the crystal dimensions along the *c*-axis remained nearly unchanged, with measured lengths of approximately 41.0 and 47.1 μm , respectively. In contrast, an expansion of about 8% was observed along the *ab*-plane, which is consistent with the changes indicated by crystallographic analysis. We prepared 60 μm size crystals, and conducted metalation under the same condition (Figure S18). We found 100 % conversion with single crystallinity. We have not been able to prepare larger single crystals than 60 μm , which makes it difficult to investigate the size limitation.

We modified the main manuscript on page 5, paragraph 2 as follows: “We also prepared 60 μm crystals and conducted the metalation under the same conditions. 100% conversion with single crystallinity was observed (Figure S18).”

Figure R1. *In situ* optical microscopy observations on two single crystals during the metalation process under visible light.

2. In Figure 4E, the slope lines are not straight but angulated.

Re: We assume that the reviewer was referring to the fact that the linear fit of the emission peaks against the polarity of protic solvents (2-propanol, ethanol, methanol, 2,2,2-trifluoroethanol, H₂O) is angulated. This is because the third point (methanol) deviates from the linear fit. This phenomenon also appears in other reported works in the same characterization (Refs. 61 and 62). These works discuss the reason for the deviation of methanol, which is due to the stronger hydrogen bonding interaction between methanol and MOFs than other protic solvents. This phenomenon is also found in our system.

We revised the manuscript to clarify this point on Page 13, paragraph 1: “Methanol would have stronger hydrogen bonding interaction with MOF-b-Cu than other protic solvents, causing it to slightly deviate from the linear fit of the emission peaks against the polarity.”^{61, 62}

Responses to Reviewer #3:

The manuscript from Wei and Horike reported the transformation of a hydrogen-bonded organic framework (HOF) to a metal organic framework (MOF) through metal metallization. The metalation time control study further revealed that a 1D axial MOF|HOF|MOF heterostructure can be achieved. The crystals exhibited interface-controlled gas diffusion and various spatially-resolved photoluminescence behaviors depending on the distribution of each component. Overall, the concept of MOF|HOF heterostructure is remarkable, and the conclusions are supported by a series of thorough characterizations. I suggest publish this work in *Nat. Commun.* after the following points are addressed.

1. Recently, Dan Li et al. reported JNU-304 in *J. Am. Chem. Soc.* 2025, 147, 16, 13711–13720. It has the same layer structure with **MOF-b-Cu**, although the packing looks different. The authors need to mention this work and discuss the difference.

Re: We found that the recently reported **JNU-304** has the same trimer topological structure, but a different AA stacking model compared with **MOF-b-Cu**. This is due to different synthesis methods. **JNU-304** was synthesized by ultrasonic reaction of 2,4,6-tris(4-aminophenyl)-1,3,5-triazine, 3,5-dimethyl-1H-pyrazole-4-carbaldehyde, and Cu_2O in 4 M aqueous acetic acid within 1 hour. **MOF-b-Cu** was synthesized via a post-synthetic metalation from **HOF-b** in Cu^+ methanol solution in 48 hours. It is interesting to have different crystal structures with the same chemical constituents, depending on the synthesis protocols.

We cited this reference (Ref. 48) and revised the manuscript to clarify this point on Page 7, paragraph 1: “**MOF-b-Cu** should have the same AB stacking model as **HOF-b**. This differs from the AA-stacking structure of a reported MOF prepared by ultrasonic reaction of 2,4,6-tris(4-aminophenyl)-1,3,5-triazine, 3,5-dimethyl-1H-pyrazole-4-carbaldehyde, and Cu_2O in 4 M aqueous acetic acid within 1 hour.⁴⁸”

2. Strong interaction between the $\text{Cu}_3(\text{pyrazolate})_3$ species are well known, so is there any possibility for **MOF-b-Cu** to have different stacking configuration with the **HOF-b**?

Re: As the reviewer suggested, the Cu-Cu interaction between the trimers is known to be strong. In previous papers about 2D $\text{Cu}_3(\text{pyrazolate})_3$ -based MOFs (*Nat. Commun.* 2024, 15, 194), we found AA to ABC transformations upon external stimulation, such as heating, changing synthesis temperature, etc. We conducted differential scanning calorimetry (DSC) and high-temperature metalation to try to observe the possible transformation from

the AB stacking configuration. DSC profile from 228 to 563 K under N₂ shows that no solid-to-solid transition, suggesting that heating in this temperature range does not lead to another form (Figures S31 and S32). The sample after metalation at 338 K exhibits the same PXRD pattern as the original **MOF-b-Cu**, indicating that the AB structure is stable in this system.

Based on these results, we modified the following main manuscript (Page 7, paragraph 2): “The AB stacking structure of **MOF-b-Cu** was intact even after heating at 563 K, or metalation at 338 K, confirmed by differential scanning calorimetry and PXRD (Figures S31 and S32).”

RESPONSE TO REVIEWERS' COMMENTS

Responses to Reviewer #1:

In the manuscript, a method for the preparation of 1D axial heterostructures by metallization strategies was developed. After the author's revision, the manuscript was further refined and supplemented. However, the current manuscript requires minor revisions. Here give some suggestions for further revision.

1. The symbol for the range between numbers is "-" instead of "-"; The font of the formula should be consistent with the body (Page 8, Line 452).

Re: We changed the "-" to "-" between numbers and modified the font of the formula to be consistent with the main text (Page 15, Line 443).

2. In the subsection "Metalation of HOF-a and HOF-b", the author mainly uses MOF-b-Cu as the object of analysis. So, what is the reason for choosing MOF-b-Cu over MOF-a-Cu, MOF-b-Ag, or MOF-b-Au? What are the advantages of MOF-b-Cu over them?

Re: We selected **MOF-b-Cu** primarily because its metalation proceeds with nearly 100% conversion from **HOF-b** within 48 hours at room temperature, which greatly facilitates mechanistic studies. In comparison, under identical conditions, the metalation conversions of **MOF-a-Cu**, **MOF-b-Ag**, and **MOF-b-Au** were 90.1%, 94.2%, and 93.2%, respectively. The complete and rapid metalation of **MOF-b-Cu** makes it an ideal system to probe the reaction pathway with minimal interference from incomplete conversion. In addition, Cu⁺ salts are significantly more cost-effective than Ag⁺ and Au⁺, making MOF-b-Cu not only experimentally more reliable but also economically advantageous.

The results and explanation were added to the "Results and Discussion" (Page 6, paragraph 1): "Under the same conditions to prepare **MOF-b-Cu**, we confirmed that 90.1%, 94.2%, and 93.2% of metalation occurred to have **MOF-a-Cu**, **MOF-b-Ag**, and **MOF-b-Au**, respectively."

3. Does this metallization method work with other types of HOF materials?

Re: Based on this work, we found that continuous pores and accessible hydrogen bonds in HOF crystals and the selection of appropriate metal salts are important factors for the occurrence of metalation. There seem to be many HOF structures that meet this requirement (Refs. 39, 40, and 41), which suggests that our strategy may be widely applicable. We revised the main manuscript on Page 9, paragraph 2: "Considering that

the reported HOFs have a variety of accessible hydrogen-bonding sites and pore structures³⁹⁻⁴¹, this strategy is expected to apply to other types of HOFs with different metal ions and anions.”